# *ABCB1* Polymorphism Is Associated with Higher Carbamazepine Clearance in Children

**DOI:** 10.3390/pediatric17010010

**Published:** 2025-01-16

**Authors:** Natasa Djordjevic, Jelena Cukic, Dragana Dragas Milovanovic, Marija Radovanovic, Ivan Radosavljevic, Jelena Vuckovic Filipovic, Slobodan Obradovic, Dejan Baskic, Jasmina R. Milovanovic, Slobodan Jankovic, Dragan Milovanovic

**Affiliations:** 1Department of Pharmacology and Toxicology, Faculty of Medical Sciences, University of Kragujevac, Svetozara Markovica 69, 34 000 Kragujevac, Serbia; jasminamilo@yahoo.com (J.R.M.); sjankovic@fmn.kg.ac.rs (S.J.); piki@fmn.kg.ac.rs (D.M.); 2Public Health Institute, Nikole Pasica 1, 34 000 Kragujevac, Serbia; jelenar.cukic@gmail.com (J.C.); dejan.baskic@gmail.com (D.B.); 3College of Professional Health Studies, Cara Dusana 254, 11 080 Belgrade, Serbia; drdragas@yahoo.com; 4Department of Pediatrics, Faculty of Medical Sciences, University of Kragujevac, Svetozara Markovica 69, 34 000 Kragujevac, Serbia; marijar9@verat.net (M.R.); sobradovic2@gmail.com (S.O.); 5Department of Surgery, Faculty of Medical Sciences, University of Kragujevac, Svetozara Markovica 69, 34 000 Kragujevac, Serbia; ivanradoskapi@gmail.com; 6Department of Internal Medicine, Faculty of Medical Sciences, University of Kragujevac, Svetozara Markovica 69, 34 000 Kragujevac, Serbia; jelenavufi@gmail.com; 7Department of Microbiology, Faculty of Medical Sciences, University of Kragujevac, Svetozara Markovica 69, 34 000 Kragujevac, Serbia

**Keywords:** *ABCB1*, genetic polymorphism, carbamazepine, population pharmacokinetics, children

## Abstract

The aim of our study was to investigate the role of *ABCB1* polymorphism in the pharmacokinetics of carbamazepine (CBZ) in children. The study enrolled 47 Serbian pediatric epileptic patients on CBZ treatment. Genotyping for *ABCB1* 1236C<T (rs1128503), 2677G<A/T (rs2032582) and 3435C<T (rs1045642) was carried out using the TaqMan method. Steady-state CBZ serum concentrations were available from our previous study, determined by high pressure liquid chromatography (HPLC). The NONMEM software and one-compartment model were used for pharmacokinetic analysis. *ABCB1* 1236C<T, 2677G<A/T and 3435C<T variations were found at the frequencies of 47.9%, 48.9% and 52.1%, respectively. The equation that described population clearance (CL) was CL (L/h) = 0.175 + 0.0403 × SEX + 0.0332 × *ABCB1* + 0.0176 × CYP1A2 + 0.000151 × DD where SEX has a value of 1 if male and 0 if female, *ABCB1* has a value of 1 if C-G-C/T-T-T and 0 if any other *ABCB1* diplotype, CYP1A2 has a value of 1 if −163A/A and 0 if −163C/C or C/A, and DD is the total CBZ daily dose (mg/day). The presence of the *ABCB1* 1236T-2677T-3435T haplotype is associated with an increased clearance of CBZ in pediatric epileptic patients.

## 1. Introduction

Carbamazepine (CBZ) is an old, well known and widely used anticonvulsant. Yet, it continues to attract the attention of the medical and scientific communities, due to the pronounced inter-individual differences it can cause in patient responses and its high propensity for drug interactions. Both seem to be the consequences of CBZ’s complicated pharmacokinetics, the features of which have been only partly explained. More precisely, drug absorption, metabolism and elimination in both adults and children have been well described for almost half a century [1,2], but its transport over the blood–brain barrier (BBB) has still not been fully clarified [3,4].

The multidrug resistance protein 1 (MDR1) is an ATP-binding cassette transporter that functions as a cellular efflux pump [5]. It is expressed at membranes, including the BBB, and affects the bioavailability and pharmacologic effects of many drugs [6,7]. However, whether the MDR1 transports CBZ is still a matter of debate, as either option has been supported by results from animal or laboratory studies [4]. Namely, it has been shown, both in vivo [8,9] and in vitro [10,11], that inhibition of the MDR1 increases intracellular CBZ concentration. On the other hand, no difference in brain penetration or anticonvulsant CBZ efficacy was observed between wildtype and MDR1-deficient mice [12,13], nor in its transportation through Caco-2 cells with extensive MDR1 activity in the presence of MDR1 inhibitors [14]. At the same time, one human study, conducted on resected brain tissue samples from drug-resistant patients undergoing epilepsy surgery, concluded that a co-administration of MDR1 inhibitor increases CBZ effectiveness [15]. In addition, one case report described a great improvement in overall seizure control in a CBZ-treated patient after CBZ was co-administered with verapamil, a MDR1 inhibitor [16]. Hence, CBZ is currently classified as a probable substrate of the MDR1 [17].

The MDR1 is encoded by the *ABCB1* (*MDR1*) gene, which consists of 29 exons set within a highly variable 209.6kb genomic region [6]. Previous studies have indicated that a number of *ABCB1* single nucleotide polymorphisms (SNPs), including synonymous variants, may result in a transporter with altered activity, expression level or substrate specificity [18]. The most common and most frequently studied *ABCB1* coding SNPs—rs1128503 (1236C>T), rs2032582 (2677G>A/T) and rs1045642 (3435C>T)—happen to be the most controversial ones. In spite of numerous attempts to identify their impact on MDR1-dependent drug transport, the overall result remains inconclusive [19]. Therefore, the aim of our study was to investigate the possible role of *ABCB1* polymorphism in the pharmacokinetics of CBZ in children.

## 2. Materials and Methods

A total of 47 patients with epilepsy, treated with CBZ for at least 4 weeks, were enrolled in the study [20]. They were recruited from the pediatric department of the Clinical Centre, Kragujevac, Serbia, with the written informed assents and consents obtained from all patients and their parents. To be included in the study, patients had to be older than 3 years of age, and diagnosed with either partial or generalized tonic–clonic epilepsy. None of the participants had contraindications for CBZ treatment [20]. The study was approved by the ethics committee at the Clinical Centre, Kragujevac, Serbia (approvals No 01-7848 and No 01/20-401), and conducted in accordance with the Declaration of Helsinki and its subsequent revisions.

Genotyping for *ABCB1* 1236C<T (rs1128503), 2677G<A/T (rs2032582) and 3435C<T (rs1045642) was carried out on SaCycler-96 (Sacace Biotechnologies, Como, Italy), with the use of the TaqMan Genotyping Master Mix 2X and the corresponding TaqMan DME genotyping Assays (Applied Biosystems, Foster City, CA, USA). For 40 patients, steady-state CBZ serum concentrations and CYP1A2 − 163C>A genotype data were available from our previous study [20].

Population pharmacokinetic analysis was performed using the non-linear, mixed-effects modeling program (NONMEM), version 7.3.0 (Icon Development Solution, Dublin, Ireland), with an ADVAN1 subroutine that describes a one-compartment model without absorption. The effects of 21 different covariates (including age, sex, body weight, total CBZ daily dose, co-therapy with valproate, CYP1A2 −163C>A genotype, as well as *ABCB1* data: genotypes, genotype groups and the most frequent diplotypes) were assessed based on the difference between the minimum of objective function (MOF) of the base and the incorporated covariate model. When a covariate was included in a univariate analysis, a decrease in MOF>3.84 (*p* < 0.05, df = 1) was considered statistically significant. To be considered as a candidate for the full model, the influence of covariates was re-tested by the process of backward deletion with an MOF difference greater than 6.6 (*p* < 0.005, df = 1). The stability and predictive performances of the final model were assessed through an internal validation process, using bootstrap analysis.

For haplotype analysis, the Hardy–Weinberg equilibrium and linkage disequilibrium calculations; the population genetic software programs Arlequin, version 3.11; (http://cmpg.unibe.ch/software/arlequin3); and Haploview, version 4.2 (https://www.broadinstitute.org/haploview/haploview), were used. Subjects were assigned to genotype groups assuming dominant and recessive genetic models, which compare carriers of at least one, and carriers of both variant alleles, respectively, with other genotypes [21]. The normality of data distribution was confirmed by Shapiro–Wilk test. Doses of CBZ were presented as adjusted per weight, and the observed drug concentrations were normalized by dose per body weight. All continuous variables were expressed as the median, interquartile range (IQR) and range. Comparisons among *ABCB1* genotypes, genotype groups and diplotypes in terms of CBZ doses and CBZ serum concentrations were made with one-way ANOVA. Statistical analyses were carried out using Statistica, version 7.1 (StatSoft, Tulsa, OK, USA), and differences at *p* < 0.05 were considered significant.

## 3. Results

There were 29 male and 18 female patients (mean age ± SD: 10.32 ± 3.06) included in the study; 40 of them were diagnosed with idiopathic epilepsy, and 7 with symptomatic epilepsy. In addition to CBZ therapy, five patients were co-treated with valproate. Other patients’ demographic and treatment-related data are presented in Table 1.

The frequencies of observed *ABCB1* alleles, genotypes and the most frequent diplotypes are presented in Table 2.

All genotype frequencies were in accordance with the Hardy–Weinberg equilibrium (*p* ≥ 0.373). Significant linkage disequilibrium was detected among all SNPs, with D’ and r2 coefficients of 0.84 and 0.67, respectively, between 1236C<T and 2677G>A/T, 0.62 and 0.33 between 2677G>A/T and 3435C<T, and 0.56 and 0.28 between 1236C>T and 3435C<T (*p* values ≤ 0.0004).

There was no significant association between any of the examined *ABCB1* genotypes, genotype groups or diplotypes, and either weight-adjusted daily dose (*p* ≥ 0.10), or dose-normalized serum concentration of CBZ (*p* ≥ 0.09). However, the tendency towards higher serum concentrations (*p* = 0.09) was observed in carriers of C-G-C/C-G-T.

Of all covariates examined by population pharmacokinetics, only six showed to be of significance for the CBZ clearance: sex, total CBZ daily dose, CYP1A2 genotype, *ABCB1* C-G-C/T-T-T and C-G-C/C-G-T diplotypes and co-therapy with valproate. The necessary additional decrease in the MOF value was not confirmed for the last two, and the final pharmacokinetic model was as follows:CL (L/h) = 0.175 + 0.0403 × SEX + 0.0332 × *ABCB1* + 0.0176 × CYP1A2 + 0.000151 × DD
where SEX has a value of 1 if male and 0 if female, *ABCB1* has a value of 1 if C-G-C/T-T-T and 0 if any other *ABCB1* diplotype, CYP1A2 has a value of 1 if −163A/A and 0 if −163C/C or C/A and DD is the total CBZ daily dose (mg/day).

A reduction in the MOF value and both inter-individual and residual variability was notably recorded in the final model, with the values of 176.807, 20.97% and 16.22%, respectively. Parameter estimates of the final model and bootstrapping analysis were similar, suggesting a good accuracy and stability of the final model (Table 3).

## 4. Discussion

In the present study, we investigated the effect of the three most important *ABCB1* coding SNPs 1236C>T, 2677G>A/T and 3435C>T on CBZ’s pharmacokinetics in pediatric epileptic patients. Our main finding was that the presence of the *ABCB1* 1236T-2677T-3435T haplotype was associated with an increased clearance of CBZ in children. To the best of our knowledge, this is the first population pharmacokinetic study to evaluate and report on the effect of *ABCB1* polymorphism on CBZ clearance.

The overall MDR1 activity depends on the level of *ABCB1* expression and the functionality of the *ABCB1*-coded protein, both being affected by *ABCB1* polymorphism [22]. The most common *ABCB1* SNPs include two synonymous and one non-synonymous variations—namely 1236C>T (Gly412Gly), 3435C>T (Ile1145Ile) and 2677G>A/T (Ala893The/Ser), respectively [6]. The frequencies of these SNPs vary by ethnicity [6,23,24], yet seem to be rather uniform among Caucasians [24,25,26,27,28,29,30,31]. In the present study, we have genotyped Serbian epileptic patients, and our results do not differ significantly from previous observations in healthy Serbs [30,31] or in other healthy Caucasian populations [24,25,26,27,28,29]. A similar distribution of *ABCB1* SNPs observed in epileptic patients and healthy subjects support a lack of association between *ABCB1* polymorphism and the risk of having epilepsy [32].

On the other hand, the available evidence, provided by systematic reviews and meta-analysis, suggests that *ABCB1’s* genetic variability, especially the silent 3435C>T variation, affects responses to antiepileptic drugs [33,34]. Silent gene variations do not alter the primary structure of the protein, but could lead to changes in protein function, due to its altering of messenger RNA splicing or protein stability, or due to linkage disequilibrium with another functional SNP, such as 2677G>A/T [35,36,37,38]. In terms of *ABCB1*, all three investigated variations, both separately, and co-existing on the same haplotype, have been associated with altered MDR1 function [39]. Yet, the results were conflicting and the effect was mostly drug-dependent [37,40,41,42,43,44,45,46]. In regard to antiepileptic treatment, observations differed both by drug and by population [34,47].

Previous CBZ-related MDR1/*ABCB1* investigations yielded contradictory conclusions on both the role of MDR in CBZ transport, and the effect of *ABCB1* genetic polymorphism on CBZ’s pharmacokinetics or efficacy. The possibility of CBZ being a substrate for the MDR1, introduced 25 years ago [48], have been both supported [8,9,49] and opposed [11,13,14,15,50,51,52] over the years. While the proposed classification renders CBZ a probable substrate of the MDR1 [17], currently there is no consensus on the subject [53]. Similarly, reports on the association between *ABCB1* polymorphism and CBZ-based therapy outcomes have been inconsistent too, and the results of these studies often depended on ethnicity [54]. Several groups have shown a more efficient transportation of CBZ [55] and lower CBZ plasma concentrations [56] in carriers of the 3435TT genotype. According to others [57,58,59,60], carriers of the 3435TT, 2677TT or 1236T-2677T-3435T haplotype more frequently suffer from CBZ-resistant epilepsy. Finally, there have been studies that have failed to detect any significant impact of *ABCB1* genetic variability on CBZ’s pharmacokinetics [57,61,62] or CBZ-based treatment outcomes [61,63,64,65,66].

In our study, population pharmacokinetic analysis predicted higher CBZ clearance in carriers of the *ABCB1* 1236T-2677T-3435T haplotype. According to the model, the effect of *ABCB1* genetic polymorphism on CBZ’s pharmacokinetics seems to be even more pronounced than the already reported effects of CBZ dosag and the presence of the *CYP1A2* −163C>A variation [20,67]. Yet, compared to other predictors, the effect of *ABCB1* genetic variability is more difficult to interpret, as neither pharmacokinetic nor pharmacogenetic-based connections between MDR1/*ABCB1* and CBZ have been firmly established yet.

Nevertheless, the most important CBZ metabolite, CBZ-10,11-epoxide (CBZ-E) [68,69], is a confirmed substrate for the MDR1 [11,70]. It has been demonstrated that CBZ-E displays equal activity as its parent compound [1,2,71]. Therefore, a previously reported association between the *ABCB1* 1236T-2677T-3435T haplotype and higher CBZ resistance [57,72] could be due to an increased MDR1-dependent efflux of CBZ-E at the BBB level. Similar to CBZ [1], CBZ-E induces several drug metabolizing enzymes [73], including the most important CBZ-metabolizing enzyme, CYP3A4 [74,75]. Therefore, we hypothesize that the increased plasma CBZ-E concentration, consequent to increased MDR1-dependent efflux at the BBB in the presence of all three variant *ABCB1* alleles, is the reason for the observed increase in CBZ (metabolic) clearance. There are, of course, other possible explanations for the results of our study, including the possibility that CBZ is a substrate of the MDR1 [8,9,48,49]; that the presence of the *ABCB1* 1236T-2677T-3435T haplotype leads to decreased MDR1-dependent CBZ transport; or that the same haplotype somehow affects MDR1 induction by either CBZ [76] or potentially CBZ-E.

In the present study, the steady-state CBZ serum concentrations were available for only 40 patients, so the sample size would most probably be too small to reveal any statistically significant effect of the examined covariates. However, our results are based on the population pharmacokinetic analysis, for which it has been demonstrated that can accurately estimate pharmacokinetic parameters based on analysis of only one blood sample, taken from as few as five subjects [77]. Therefore, we believe our conclusions are sound. Nevertheless, performing a replication study involving more participants and more than one blood sample would be advisable to confirm our findings. In addition, it should be noted that the production of CBZ-E and its contribution to CBZ anticonvulsant effects seem to be more pronounced in children [76,77,78], so the association between *ABCB1* polymorphism and higher CBZ clearance observed in our study might not be present in the adult population.

In addition to the *ABCB1* genotype, the present study confirmed that the male sex, the *CYP1A2* −163A/A genotype and higher CBZ daily dosag are associated with increased CBZ clearance [20]. While the influence of sex could be attributed to the inhibiting effect of female sex hormones on CYP1A2 activity [79], the presence of the *CYP1A2* −163A/A genotype most probably promotes the enzyme inducing effect of CBZ [20]. The importance of total daily doses for CBZ clearance has been long recognized [67], but in our study, when compared to other significant predictors, its impact was shown to be of less importance. This lower-than-expected effect of the drug’s dose has been previously reported [20], and the possible explanations include a potential bias due to concentration-dependent dose adjustments [80], as well as the dose-dependent autoinduction of CBZ metabolism [67].

It should be noted that, due to the wide inter-individual variability in response to treatment, interactions with other drugs and narrow therapeutic range, CBZ represents an excellent candidate for therapeutic drug monitoring (TDM) [81,82]. Previous studies have confirmed that routine TDM provides clinicians with valuable guidance for better CBZ dosing management, in order to minimize side effects and optimize treatment efficacy [83,84,85]. However, due to the metabolic autoinduction of CBZ, the time it takes to reach a steady state at the initiation of the therapy might be prolonged by up to several weeks [86], causing a delay in accurate serum concentration measurements, thereby increasing the risk of an inadequate dosing regimen. Therefore, an optimization of CBZ-based treatments calls for both TDM and pharmacogenetics to complement each other in practicing evidence-based personalized medicine.

In conclusion, our study reveals a higher CBZ clearance in the presence of the *ABCB1* 1236T-2677T-3435T haplotype in pediatric epileptic patients, supporting the potential of personalized medicine in optimizing epilepsy treatment. The reason for the observed association, as well as whether the same can be expected in adults and in other ethnic groups, remains to be explored.

## Figures and Tables

**Table 1 pediatrrep-17-00010-t001:** Patient characteristics and medication data.

Patient Characteristics	
Body weight (kg)	
Median (IQR)	39.0 (31.5–48.5)
Range	17.0–65.0
Age (years)	
Median (IQR)	11 (8–12)
Range	4–17
Medication data	
Carbamazepine daily dose (mg/kg)	
Before dose adjustment	
Median (IQR)	15.38 (12.06–18.94)
Range	6.90–24.24
After dose adjustment	
Median (IQR)	16.33 (12.50–18.79)
Range	6.90–24.24
Carbamazepine serum concentration (mg/L)	
Before dose adjustment	
Median (IQR)	6.40 (5.25–7.20)
Range	3.52–9.93
After dose adjustment	
Median (IQR)	6.23 (5.86–7.73)
Range	2.71–10.58

**Table 2 pediatrrep-17-00010-t002:** Allele, genotype and diplotype frequencies of *ABCB1* in epileptic patients on carbamazepine treatment.

		Observed Frequency	95% Confidence Interval
**Allele**			
rs1128503 (1236T)		0.479 (45/94)	0.381, 0.579
rs2032582 (2677T/A)		0.489 (44/94)	0.391, 0.589
rs1045642 (3435T)		0.521 (49/94)	0.421, 0.619
**Genotype**			
rs1128503 (1236C>T)	CC	0.234 (11/47)	0.135, 0.375
	CT	0.574 (27/47)	0.433, 0.705
	TT	0.191 (9/47)	0.103, 0.329
rs2032582 (2677G>T/A)	GG	0.255 (12/47)	0.152, 0.397
	GT	0.511 (24/47)	0.373, 0.647
	GA	0.000 (0/47)	0.000, 0.092
	TT	0.191 (9/47)	0.103, 0.329
	AT	0.043 (2/47)	0.005, 0.152
	AA	0.000 (0/47)	0.000, 0.092
rs1045642 (3435C>T)	CC	0.149 (7/47)	0.072, 0.281
	CT	0.660 (31/47)	0.516, 0.778
	TT	0.191 (9/47)	0.103, 0.329
**Diplotype 1236-2677-3435** *****			
C-G-C/C-G-C		0.064 (3/47)	0.016, 0.180
C-G-C/C-G-T		0.128 (6/47)	0.057, 0.257
C-G-C/T-T-T		0.362 (17/47)	0.240, 0.505
T-T-T/T-T-T		0.128 (6/47)	0.057, 0.257

* other diplotypes were deemed too rare to yield any significant association, and thus were not included in statistical and population pharmacokinetics analyses.

**Table 3 pediatrrep-17-00010-t003:** The final model parameter estimates.

	NONMEM		BOOTSTRAP ANALYSIS	
**Parameter**	**Estimate**	**95% CI ***	**Estimate**	**95% CI ^‡^**
CL/F (L/h)	0.175	0.14–0.21	0.170	0.16–0.19
SEX	0.0403	0.0324–0.0482	0.0401	0.0269–0.0537
DD (mg/L)	0.000151	0.000097–0.000205	0.000154	0.00011–0.00019
CYP1A2	0.0176	0.0133–0.0219	0.0181	0.0163–0.0199
ABCB1	0.0332	0.0235–0.0429	0.0329	0.023–0.0367
ω^2^_CL_	0.0361	0.0216–0.0506	0.0385	0.0308–0.0415
σ^2^	0.025	0.0191–0.0309	0.0257	0.022–0.0294

* (Estimate) ± 1.96 × (standard error of the estimate). ^‡^ 2.5th and 97.5th percentile of the ranked bootstrap parameter estimates. ω^2^_CL_—Interindividual variance of CL. σ^2^—Residual variance.

## Data Availability

The raw data supporting the conclusions of this article will be made available by the authors on request.

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
