# Peer review of "ABCB1 Polymorphism Is Associated with Higher Carbamazepine Clearance in Children"

_pediatrrep, 2025, doi:10.3390/pediatric17010010_

Round 1
Reviewer 1 Report
Comments and Suggestions for Authors
The authors undertook the difficult task of assessing the ABCB1 polymorphism in pediatric patients treated with carbamazepine. The study included 47 patients of Serbian nationality who were treated with carbamazepine for epilepsy. The authors observed that one of the haplotypes studied was associated with increased creatinine clearance in pediatric patients.
I believe that the work is very well written. The introduction introduces the reader to the problem, the Material and Methods section precisely describes the methodology used, and the Results section presents the data in a clear manner. The work is also well discussed.
However, I have a few comments to introduce, which I hope will improve the quality of the publication. In the introduction, as well as in the summary, the authors mention carbamazepine concentrations in blood serum. Therapy monitored by the concentration of the drug, in this case carbamazepine, has been a common practice since the 1990s. Carbamazepine concentration determinations, as an older generation antiepileptic drug, have always been mandatory.
I ask the authors to address this issue in the introduction, preferably in the form of a subsection on monitored therapy. It would be worth emphasizing that monitoring serum carbamazepine concentration is not only standard, but also allows for better dosing management in order to minimize side effects and optimize treatment efficacy. There are publications describing therapeutic drug monitoring of carbamazepine: a 20-year observational study. This may be particularly important in the context of work, where the relationship between genetic polymorphism and drug metabolism plays a key role.
I would recommend that the authors consider including a short one-paragraph review of the use of monitored therapy in carbamazepine treatment, including the relationship with pharmacogenomics. Such a subsection could be included in the introduction of the work and would enhance its value, and the work is to be a source not only for basic science researchers but also clinicians. It would be good to indicate more clearly in the conclusions the need for TDM, especially in these patients.
Reviewer 2 Report
Comments and Suggestions for Authors
A simple, straight-forward study of the effect of ABCB1 diplotypes on carbamazepine clearance in children. Presentation does suffer however from a number of weaknesses, summarized below:
1. The authors need to add some information on patient recruitment, demographics, clinical pharmacology etc., in materials and methods. Referral to their previous work (ref. 20) is simply not enough, in my view at least. Carbamazepine doses and clearance values in particular should be there for the reader to see directly, without having to look into another publication.
2. In Table 1, individual heterozygote frequencies for the three SNPs are 0.511, 0.574, and 0.660, yet the CGC/TTT diplotype has a frequency of 0.362. Since the three SNPs are known to be tightly linked, it would be a good idea to calculate measures of linkage disequilibrium (i.e., D’ or r2) between any of the three loci.
3. In Results, lines 133-134, p-values of 0.450 or 0.250 are hardly worth mentioning to conclude that higher doses and lower serum concentrations (not shown!) are associated with the CGC/TTT diplotype.
4. Please add p-values in Table 2; also, R2 (what percentage of total variation in CBZ clearance does the linear regression model explain?).
5. According to the model, sex has the higher effect on CBZ clearance, followed by ABCB1 diplotype, CYP1A2 genotype and, finally, daily dose, which appears to exert a very small effect indeed (one additional mg per day increases CL by 151 μL per hour!). Yet the authors chose to not comment on the relative effect of all the independent variables included in the model and focus exclusively on ABCB1, which makes little sense. Also, in absence of real overall clearance values, it is almost impossible to decide on the real effect of those variables (i.e., according to the model, carriers of the CGC/TTT diplotype should display increased CBZ clearance by 33 mL per hour compared to the other diplotypes, but what does that really mean?).
Reviewer 3 Report
Comments and Suggestions for Authors
The manuscript titled “ABCB1 polymorphism is associated with higher carbamazepine 2 clearance in children” explores the impact of ABCB1 polymorphism on the pharmacokinetics of carbamazepine (CBZ) in children with epilepsy. The study, conducted on 47 Serbian pediatric epileptic patients, used genotyping for specific ABCB1 SNPs (1236C>T, 2677G>A/T, 3435C>T) and pharmacokinetic modeling to analyze the influence of genetic variation on CBZ clearance. The authors reported that the ABCB1 1236T-2677T-3435T haplotype was associated with increased clearance of CBZ in pediatric patients. Other genetic and demographic factors (e.g., CYP1A2 polymorphism, sex) also influenced CBZ pharmacokinetics. Clearance (CL) was influenced by sex, total CBZ dose, CYP1A2 genotype, and ABCB1 diplotypes. Model validation using bootstrap analysis confirmed the robustness of the results. The authors indicated that higher CBZ clearance in children with the ABCB1 1236T-2677T-3435T haplotype may have clinical implications, especially for dosing strategies.
The following is this reviewer’s comments:
This study addresses an important pharmacogenetic question and utilizes a rigorous population pharmacokinetic approach. The findings highlight the potential of personalized medicine in optimizing epilepsy treatment.
Although, the small sample size reduces the statistical power and generalizability of the findings. Future research with larger cohorts is recommended.
Round 2
Reviewer 1 Report
Comments and Suggestions for Authors
The authors have made appropriate corrections to the manuscript. I believe that the manuscript can now be considered for publication.
Reviewer 2 Report
Comments and Suggestions for Authors
I have no further comments.